# Sesquiterpene-Loaded Co-Polymer Hybrid Nanoparticle Effects on Human Mast Cell Surface Receptor Expression, Granule Contents, and Degranulation

**DOI:** 10.3390/nano11040953

**Published:** 2021-04-08

**Authors:** Narcy Arizmendi, Hui Qian, Yiming Li, Marianna Kulka

**Affiliations:** 1Nanotechnology Research Centre, National Research Council Canada, Edmonton, AB TG6 2M9, Canada; Narcy.Arizmendi@nrc-cnrc.gc.ca (N.A.); Hui.Qian@nrc-cnrc.gc.ca (H.Q.); 2Department of Medical Microbiology and Immunology, Faculty of Medicine, University of Alberta, Edmonton, AB T6G 2S2, Canada; 3School of Pharmacy, Shanghai University of Traditional Chinese Medicine, Shanghai 250014, China; 0000001968@shutcm.edu.ca

**Keywords:** allergic inflammation, nanoparticles, eremophilane-type sesquiterpenes, cryoTEM

## Abstract

Biodegradable polymeric nanoparticles (NPs) such as poly(lactic-co-glycolic acid) (PLGA) and polyvinyl alcohol (PVA) have been used as drug delivery systems for natural and synthetic compounds and are designed to control the loading and release of biodegradable materials to target cells, tissues, and organs. Eremophilane-type sesquiterpenes have anti-inflammatory properties but are lipophilic, cytotoxic, and not biocompatible with many cells. To determine whether biodegradable PLGA/PVA could improve the biocompatibility of sesquiterpenes, sesquiterpene-loaded NPs were synthesized and their effects on human mast cells (LAD2), the major effector cells of allergic inflammation, were determined. NPs composed of PLGA/PVA and two types of sesquiterpenes (fukinone, PLGA/PVA-21 and 10βH-8α,12-epidioxyeremophil-7(11)-en-8β-ol, PLGA/PVA-22) were produced using a microfluidic synthesis method. The NPs’ size distribution and morphology were evaluated by dynamic light scattering and cryogenic transmission electron microscopy (TEM). PLGA/PVA-21 and PLGA/PVA-22 were 60 to 70 nm and were readily internalized by LAD2 as shown by flow cytometry, fluorescence microscopy, and TEM. While unencapsulated sesquiterpenes decreased LAD2 cell viability by 20%, PLGA/PVA-21 and PLGA/PVA-22 did not alter LAD2 viability, showing that encapsulation improved the biocompatibility of the sesquiterpenes. PLGA/PVA-21 and PLGA/PVA-22 decreased the expression of genes encoding the subunits of the high affinity immunoglobulin E receptor (*FcεR1α*, *FcεR1β*, *FcεR1γ*) and the stem cell factor receptor (*Kit,*), suggesting that hybrid NPs could alter mast cell responses to antigens and shift their maturation. Similarly, PLGA/PVA-21 and PLGA/PVA-22 inhibited tryptase expression but had no effect on chymase expression, thereby promoting a shift to the tryptase-positive phenotype (MC_T_). Lastly, PLGA/PVA-21 and PLGA/PVA-22 inhibited mast cell degranulation when the LAD2 cells were activated by IgE crosslinking and FcεRI. Overall, our results suggest that PLGA/PVA-21 and PLGA/PVA-22 alter human mast cell phenotype and activation without modifying viability, making them a more biocompatible approach than treating cells with sesquiterpenes alone.

## 1. Introduction

Drug-delivery of lipophilic payloads to specific intracellular compartments is particularly challenging. First, the delivery vehicle must allow for the stable incorporation and loading of the constituent components, including the target payload. Second, the delivery vehicle must possess the necessary chemical composition to release the payload in the target organelle or cellular compartment. When targeting complex immune cells such as granulocytes (basophils and mast cells), these conditions are especially challenging because these cells contain several functionally distinct intracellular structures with few distinguishing biomolecular targets.

Biodegradable polymeric NPs provide controlled and sustained release properties; their subcellular size and biocompatibility have increasingly been used for their potential targeting of tissues and cells. Poly(lactic-co-glycolic acid) (PLGA) is a copolymer of lactic and glycolic acid that has been approved by the US Food and Drug Administration (FDA) [1] and the European Medicines Agency (EMA) [2] for use in food applications due to its biocompatibility and biodegradability, and it is widely used for biomedical applications, particularly in the delivery of lipophilic compounds in aqueous environments. Recently, the FDA has approved a few nanotechnology-based drug formulations for clinical applications, and nano approaches continue to be important in the development of smarter and more powerful drug formulations. In this context, PLGA is ideal for the introduction of payloads to cellular endosomes and endolysomes since it undergoes hydrolysis of its ester linkages and the production of its monomers further creates an autocatalytic environment for complete degradation. Polyvinyl alcohol (PVA) is also highly biocompatible [3], and is used in suspension polymerizations [4] and in surfactant-free compositions to increase nanoparticle stability [5,6,7].

The objective of this study was to create a controlled-release nanoparticle capable of delivering a lipophilic molecule into a human granulocyte, thereby changing its function. We chose human mast cells as a model system, specifically the human mast cell line LAD2. Mast cells are complex inflammatory immune cells that contain several types of specialized intracellular structures such as mitochondria, free ribosomes, intermediate filaments, lipid bodies, endo-lysosomes, a monolobed nucleus [8], and lysosome-like secretory granules that contain preformed bioactive constituents including histamine, cytokines, lysosomal hydrolases, serglycin proteoglycans, and mast cell-restricted proteases [9,10]. Mast cells are activated through their surface receptors, which include the high affinity immunoglobulin E (IgE) receptor (FcεRI) that recognizes and binds to allergen-specific IgE. Allergen crosslinking of FcεRI initiates a signaling cascade that ultimately causes a process called degranulation, whereby the pro-inflammatory mediators stored in the granules are released and activate the process of allergic inflammation [11].

We designed a PLGA/PVA hybrid copolymer using a microfluidic fabrication process that allowed for highly reproducible biodegradable copolymer NPs, smaller than 100 nm in size, to target mast cell intracellular structures. In addition, we loaded these copolymers with two lipophilic eremophilane-type sesquiterpenes (SQ), which we have previously shown to modify dendritic cell function, inhibiting the maturation and activation of bone marrow-derived dendritic cells [12,13]. We hypothesized that these SQ would also modify human mast cell functions and that incorporation into the hybrid polymer delivery system would further amplify their biological effects.

## 2. Materials and Methods

### 2.1. Materials

Resomer RG 502H (PLGA, Poly (D, L-lactide-co-glicolide, ratio 50:50, MW 10–12 kDa), polyvinyl alcohol (PVA; 9–10 kDa, 80% hydrolyzed), and the laser grade fluorescent dye coumarin-6, [3-(2-benzothiazolyl)-7-(diethyl amine) coumarin] were purchased from Sigma-Aldrich (Oakville, ON, Canada). Culture cell reagents were purchased from Life Technologies (Burlington, ON, Canada). All chemicals were of the highest grade commercially available.

### 2.2. Plant Material

Bicyclic sesquiterpenes fukinone (SQ21), and 10βH-8α,12-epidioxyeremophil-7(11)-en-8β-ol (SQ22) were isolated and purified from rhizome of *Petasites tatewakianus* as previously described (12). Briefly, 10 kg of *P. tatewakianus* were extracted three times (2 h each) under reflux in 95% ethanol. The solvent was evaporated under reduced pressure to yield a crude extract, which was suspended in water and then fractionated by liquid–liquid partition sequentially with petroleum ether, chloroform, and n-BuOH to afford chloroform-soluble and BuOH-soluble fractions. The chloroform-soluble fraction of the roots of *P. tatewakianus* was subjected to column chromatography on silica gel (200 to 300 mesh), eluting with a gradient of petroleum ether/AcOEt. Further purification by column chromatography using silica gel, Sephadex LH-20, and ODS yielded sesquiterpenes SQ-21 and SQ-22 [14,15].

### 2.3. Cell Culture

The LAD2 (Laboratory of Allergic Diseases 2) human mast cell line was a generous gift of Dr. Kirshenbaum (Laboratory of Allergic Diseases, Bethesda, MD, USA). The LAD2 cells were cultured in StemPro-34 SFM medium (Life Technologies, Burlington, ON, Canada) supplemented with 2 mM L-glutamine, 100 U/mL penicillin, 50 µg/mL streptomycin (Life Technologies, Burlington, ON, Canada), and 100 ng/mL human stem cell factor (hSCF; Peprotech, Rocky Hill, NJ, USA) defined as complete media. Cell suspensions were seeded at a density of 0.1 × 10^6^ cells/mL and maintained at 37 °C and 5% CO_2_, and periodically tested for the expression of CD117 and FcεRI by flow cytometry analysis using a CytoFlex flow cytometer (Beckman Coulter, Brea, CA, USA). The cell culture was hemi-depleted every week with fresh medium.

### 2.4. PLGA/PVA Nanoparticle Formulation, Synthesis and Characterization

Nanoparticles were prepared using PLGA, 5 mg/mL in acetonitrile HPLC grade (EMD Millipore, Etobicoke, ON, Canada)/PVA 2.0% in water (PLGA/PVA). The NPs were fabricated using a NanoAssemblr Benchtop platform instrument and microfluidic chip cartridges (Precision Nanosystems Inc. Vancouver, BC, Canada), according to the manufacturer’s method, using the following parameters: 2 mL volume, flow rate ratio 1:1 (aqueous to acetonitrile), total flow rate 8 mL/min. The NPs were loaded with 25 μM of the bicyclic SQs fukinone (PLGA/PVA-21), or 25 μM 10βH-8α, 12-epidioxyeremophil-7(11)-en-8β-ol (PLGA/PVA-22); fluorescent label coumarin-6 (0.5% in acetone) was included as a positive loading control (PLGA/PVA-C6). PLGA/PVA were also included as negative controls. The NP formulations were diluted in PBS (pH 7.4, Ca^2+^ and Mg^2+^-free, Life Technologies, Burlington, ON, Canada), dialyzed against PBS, and centrifugally filtrated three times according to Precision Nanosystems Inc., using Amicon Ultra centrifugal filters (Sigma Aldrich Canada, Oakville, ON, Canada) with a low-binding regenerated cellulose membrane and a nominal MW cutoff of 10 kDa, followed by a final overnight dialysis. The NP formulations were sterile-filtered using 0.2 μm syringe filters (VWR International, Mississauga, ON, Canada).

Hydrodynamic sizes and polydispersity indexes (PDI) of the nanoparticles were analyzed post-dialysis as follows: the NP formulations were diluted (1/100 *v*/*v*) in PBS and deposited in polystyrene cuvettes (Sarstedt, Montréal, QC, Canada); the hydrodynamic diameter (Dh) and PDI were evaluated by dynamic light scattering (DLS) using a Malvern ZetaSizer Nano ZS (Malvern Instruments, Malvern Instruments, Malvern, UK) set to 25 °C and a measurement angle of 173°. The reported NP formulation values are means ± SEM of at lowest three individual measurements for each sample (Table 1). The size and morphology of the NPs were revealed in cryogenic transmission electron microscopy (cryo-TEM), JEOL2200FS TEM with an in-column Omega energy filter running at a 200 kV accelerating voltage. The cryo-TEM specimen of NPs in suspension were prepared using the plunge-freezing method. Briefly, 4 µL of NP solution was deposited on perforated carbon film supported TEM grids; excess solution was blotted away from TEM grids using filter paper, then the TEM grids were rapidly plunged into liquid ethane at −180 °C (Leica EM GP2 Plunge Freezer). The cryo-TEM specimens were transferred to a cryogenic TEM holder and kept at −180 °C for imaging in TEM. A 10 eV energy slit and defocusing were applied to enhance the contrast of bright field cryo-TEM images of NPs embedded in vitrified aqueous solution. The low-dose imaging mode was used to mitigate the electron beam damage and the electron beam dose for each cryo-TEM image was about 2 e^−^/Å^2^. The percentage of SQ encapsulated in the PLGA/PVA nanoparticles was measured by high-performance liquid chromatography (HPLC), using an Agilent 1260 HPLC (Agilent, Santa Clara, CA, USA) unit equipped with a UV-Vis detector. A Zorbax StableBound 80 Å C18, 4.6 × 250 mm, 5 µm HPLC column (Agilent, Santa Clara, CA, USA) was used as the stationary phase with an acetonitrile:water:TFA (50:50:0.1%) mobile phase. PLGA/PVA-21 and PLGA/PVA-22 were dissolved in 1 mL acetonitrile and filtered with a 0.45 µm polyvinylidene fluoride (PVDF) syringe filter for HPLC analysis. The column effluent was detected with a flow rate of 1 mL/min, and a wavelength of at 210 nm and 254 nm, with a retention time for SQ 21 and SQ 22 of 3.049 min, and 3.014 min respectively. The SQ encapsulation efficiency was calculated in the aqueous phase as follows:% encapsulation efficiency = Total SQ-loaded SQ/Total SQ × 100
where the total SQ is the amount of SQ initially added and loaded SQ is the amount of encapsulated within the PLGA/PVA nanoparticles.

### 2.5. Cytotoxic Analysis of PLGA/PVA Nanoparticle Formulation

LAD2 cells were collected and suspended at 0.2 × 10^6^ cells/mL in complete media and treated with 12.5, 25, and 50 μM of SQs SQ21, SQ22, or with NP formulations PLGA/PVA, PLGA/PVA-21, PLGA/PVA-22, and PLGA/PVA-C6 (1/100 *v*/*v*, lower NP formulations were not detected, thus we use 1/100 throughout this study), and incubated for 1, 3, 24, 48, or 72 h at 37 °C and 5% CO_2_; untreated cells were included as negative controls. Cell viability was assessed using a XTT-based proliferation assay (Roche Diagnostics GmbH, Mannheim, Germany), where the XTT reagent and electron coupling reagent mixture (sodium 3′-[1-phenylaminocarbonyl]-3,4-tetrazolium-bis (4-methoxy-6-nitro) benzene sulfonic acid hydrate as labeling reagent, and PMS (N-methyl dibenzopyrazine methyl sulfate) as electron coupling reagent) were added to the cells and incubated at 37 °C and 5% CO2 for an additional 24 to 72 h. The cell viability was analyzed using a Synergy H1 microplate reader (Biotek Instruments, Inc., Winooski, VT, USA) and expressed as the metabolic activity rate (%) of LAD2-treated cells relative to untreated control cells.

### 2.6. Intracellular Analysis of LAD2 Cells Treated with NP by Electron Microscopy

For direct NP internalization, LAD2 cells were treated for 3 and 24 h at 37 °C and 5% CO_2_ with (1/100 *v*/*v*) PLGA/PVA-21, PLGA/PVA-22, or PLGA/PVA; untreated cells were included as controls. LAD2-treated cells were collected, centrifuged at 230× *g*, and fixed overnight with 2% paraformaldehyde/2.5% glutaraldehyde (Electron Microscopy Sciences, Hatfield, PA, USA) in 0.1M phosphate buffer pH 7.2 (Sigma-Aldrich CAN, Oakville, ON, Canada). Fixed LAD2 cells were immersed in 1% osmium tetroxide (Electron Microscopy Sciences, Hatfield, PA, USA) at room temperature for 1 h before the sequential dehydration with ethanol of 50%, 70%, 95%, and 100%. The dehydrated cells were embedded in LR white resin (Electron Microscopy Sciences, Hatfield, PA, USA), and cured at 55 °C for 12 h. The ultrathin sections with 100 nm thickness, sliced by an ultra-microtome, were placed on formvar-coated copper TEM grids double-stained with uranium acetate and lead citrate. Bright field TEM images of LAD2 cell thin sections were obtained on JEOL2200FS TEM.

### 2.7. Nanoparticle Cellular Internalization and Receptor Expression Analysis by Flow Cytometry

In order to assess nanoparticle cell internalization, LAD2 cells were seeded in 24-well plates (0.2 × 10^6^ cells/mL) in growth medium, and were incubated with (1/100 *v*/*v*) NP containing the fluorescent dye coumarin-6 (PLGA/PVA-C6) for 24 h at 37 °C and 5% CO_2_, after which cells were collected, washed twice with PBS, and fixed with 5% formalin neutral buffered solution (Sigma-Aldrich, Oakville, ON, Canada), for 15 min at RT followed by 3% BSA/PBS for 10 min on ice; cells were washed and resuspended in 0.1% BSA/PBS, and analyzed on a CytoFlex flow cytometer (Beckman Coulter, Brea, CA, USA) using a 488 nm excitation and 525/40 emission filters. Untreated cells, and PLGA/PVA NPs were included as controls. The results were expressed as the median of fluorescence intensity (MFI) ± SEM; values obtained were analyzed using FlowJo v.10 software (TreeStar, Ashland, OR, USA) and compared to control values; changes in MFI were determined as nanoparticle cell internalization.

Moreover, LAD2 cells were incubated with PLGA/PVA-21 or PLGA/PVA-22 for 24 h at 37 °C and 5% CO_2_ in growth medium, washed twice with 0.1% BSA-PBS, and incubated for 1 h in the dark at 4 °C with anti-human FcεRIα-APC (0.06 µg, eBioscience, Invitrogen Carlsbad, CA, USA), anti-human c-Kit-PE (0.125 µg, eBioscience, Invitrogen Carlsbad, CA, USA), anti-human MrgprX2-PE (0.025 µg, Biolegend, San Diego, CA, USA), or anti-human tryptase antibody clone G3 (0.05 µg, Chemicon, EMD Millipore, Burlington, MA, USA), followed by 45 min incubation with goat anti-mouse Alexa Fluor 647 (0.5 µg, ThermoFisher Scientific, Waltham, MA, USA) or anti-human chymase clone B7-biotin (1:2000 dilution, MAB1254B, Chemicon, EMD Millipore, Burlington, MA, USA), followed by 45 min incubation with streptavidin-APC (0.01 µg, Southern Biotech, Birmingham, AL, USA), washed twice, and fixed as above. Isotype controls include mouse IgG1κ-PE, mouse IgG2bκ-PE and -APC, Armenian hamster IgG-APC (eBioscience, Invitrogen Carlsbad, CA, USA), and mouse IgG1 Alexa Fluor 647 (R&D Systems, Minneapolis, MN, USA). Receptor expression was analyzed as MFI ± SEM as described above.

### 2.8. Nanoparticle Cellular Internalization Analysis by Fluorescence Microscopy

Nanoparticle cell internalization was also analyzed on 5 × 10^4^ LAD2 cells treated with PLGA/PVA-C6 (1/100 *v*/*v* in complete media), and incubated for 24 h at 37 °C and 5% CO_2_; cells were collected and washed in PBS. The LAD2 cells were cytospun (Cytospin 4, Thermofisher Scientific, Waltham, MA, USA) onto glass slides at 500× *g* for 5 min, cells were fixed with 2% paraformaldehyde for 15 min, and washed twice with PBS for 5 min each, and glass slides were air-dried, followed by the addition of ProLong™ Gold antifade mountant with DAPI (Life Technologies, Carlsbad, CA, USA) and a cover slide. PLGA/PVA-C6 cell internalization was evaluated by changes in fluorescence intensity compared to untreated cells on an inverted fluorescence microscope (IX81, Olympus Canada Inc., Richmond Hill, ON, CAN), using an FITC filter.

### 2.9. Real-Time PCR

LAD2 cells were incubated with PLGA/PVA-21 or PLGA/PVA-22 NPs (1/100 *v*/*v* in complete cell media) for 6 h at 37 °C and 5% CO_2_, PLGA/PVA and untreated cells were included as controls. The LAD2-treated cells were collected, centrifuged at 230× *g*, and washed once with PBS; cell pellets were maintained in RNA for further RNA isolation and analysis. Total RNA was isolated using a RNeasy Plus Kit (Qiagen, Hilden, DEU) according to the manufacturer’s protocol, and 100 ng RNA were used to synthetize cDNA by M-MLV reverse-transcriptase (Invitrogen, Waltham, MA, USA) according to the manufacturer’s instructions. The gene expression levels of *FcεR1α*, *FcεR1β*, *FcεR1γ*, *Kit*, *MrgprX2* receptors, and *TPSAB1* (*tryptase*) and *CMA1* (*chymase*) enzymes were analyzed by a StepOnePlus real-time PCR system (Applied Biosystems, Thermofisher Scientific, Waltham, MA) with gene-specific primers and probe sets (Integrated DNA Technologies, Coralville, IA). PrimeTime^®^ qPCR oligonucleotides and probes were obtained from Integrated DNA Technologies (IDT, Coralville, IA, USA) and are shown in Table 2. The PCR mixture (20 µL total volume) consisted of the template, primer, and probes for each gene, and the PrimeTime^®^ Gene expression Master Mix (IDT, Coralville, IA, USA). Real-time PCR amplifications were carried out as follows: initial denaturation cycle at 50 °C for 2 min, and incubation at 95 °C for 10 min, followed by 40 cycles of denaturation at 95 °C for 15 s, and annealing and elongation at 60 °C for 1 min, with a final extension at 97 °C for 10 min. Results were analyzed by the 2-ΔΔCT method using HPRT1 to normalize the gene expression. Data were obtained from three independent measurements performed in duplicate and reported as a ratio of the gene of interest vs. HPRT1.

### 2.10. Degranulation Assay

LAD2 cells were washed with 0.4% BSA-HEPES buffer and suspended at 0.5 × 10^6^ cells/mL. A total of 50 µL of cell suspension were added to each well of a 96-well plate, incubated with of SQ21 (25 μM), or SQ22 (25 μM), or the nanoparticle formulations, PLGA/PVA, PLGA/PVA-21, PLGA/PVA-22 for 30 min at 37 °C and 5% CO_2_; as a degranulation control, LAD2 cells were stimulated with 1 µg/mL compound 48/80 (c48/80; Sigma-Aldrich Canada, Oakville, ON, Canada), for 30 min at 37 °C and 5% CO_2_.

For IgE-dependent activation assays, cells were sensitized overnight at 37 °C and 5% CO_2_ with 0.5 µg/mL biotinylated IgE, washed twice with 0.4% BSA-HEPES, suspended at a cell density of 0.5×10^6^ cells/mL, and incubated with SQ21, SQ22, PLGA/PVA, PLGA/PVA-21, or PLGA/PVA-22 for 3 h at 37 °C and 5% CO_2_, and then challenged with 0.1 µg/mL streptavidin for 30 min at 37 °C and 5% CO_2_. The treated LAD2 cells were centrifuged at 230× *g* for 5 min, and cell-free supernatants were collected in a different 96-well plate; cell fractions were resuspended and lysed with 0.1% Triton X-100. β-hexosaminidase release was quantified by the hydrolysis of p-nitrophenyl-N-acetyl-β-D-glucosamine (Sigma-Aldrich Canada, Oakville, ON, Canada) in 0.1 M sodium citrate buffer (pH 4.5) and analyzed using a Multiskan Ascent plate reader. Results are reported as the percentage of intracellular β-hexosaminidase released into the medium after correction for spontaneous release.

### 2.11. Statistical Analysis

All experiments were conducted in duplicates and repeated at least three separate times independently. The results are means ± standard error of the mean (SEM) of the three experiments. The data were analyzed using by Graph Pad Prism statistical software v.8 (Graph Pad, San Diego, CA, USA). One-way ANOVA with post hoc analysis and Tukey’s multiple comparison tests were used to determine differences among treatment groups. The significance level was set at *p* < 0.05.

## 3. Results

### 3.1. Effects of Eremophilane-Type SQs on Human Mast Cells

Eremophilane-type SQs used in these studies are bicyclic terpenes and are referred to as SQ 21 (fukinone, SQ21, Figure 1A), and SQ 22 (10βH-8α, 12-epidioxyeremophil-7(11)-en-8β-ol, SQ22, Figure 1B). Our previous work had shown that some forms of eremophilanes can inhibit the viability of a rat basophilic cell line at concentrations above 25 μM [16]. To determine whether SQ21 or SQ22 were biocompatible with human LAD2 cells, we measured the metabolic activity of human LAD2 mast cells exposed to different concentrations (12.5, 25 and 50 μM) of SQ21 (Figure 1C), or SQ22 (Figure 1E) for 1, 3, 48, or 72 h. LAD2 cell metabolism decreased to 76.6 ± 10%, 1 h (checkered bars) after exposure to 25 µM of SQ21. However, the metabolic activity of these cells recovered back to 89 ± 3.5% 72 h post-treatment (downward diagonal bars) when compared to untreated control cells. At 24h post-treatment, LAD2 reached a maximum metabolic cell activity (89 ± 3.5%, Figure 1D) in all tested concentrations, remaining unaffected at the latter time points. In comparison, LAD2 cell metabolism after 1 h treatment with 25 µM of SQ22 (Figure 1E) decreased to 76 ± 10.5% (checkered bars), which improved to 84.4 ± 6.6% 3 h post-treatment (open bars), reaching up to 78.7 ± 13.3% 24 h later (Figure 1F), maintaining this activity for up to 72 h in all tested concentrations. LAD2 cell metabolic activity observed with both compounds at 25 μM persisted after the 3 h treatment; therefore, we used 25 μM of both SQ21 and SQ22 for further analysis. Untreated cells were included as negative controls in all tested conditions.

### 3.2. PLGA/PVA and Eremophilane-Type SQs Generated Uniform Nanoparticles

The PLGA polymer was chosen in this study because of its biodegradable properties, which, after degradation by the citric acid cycle, releases carbon dioxide and water. These polymers have proven to be an excellent drug carrier, functioning as drug delivery vehicles in biomedical applications such as vaccination, cancer treatment, inflammation, and other chronic diseases. We postulated that a hybrid PLGA/PVA structure would be compatible with SQs and create a biodegradable nanoparticle suitable for use as a drug delivery system. We prepared PLGA/PVA NPs (PLGA/PVA), encapsulating 25 μM of SQ21 (PLGA/PVA-21), or SQ22 (PLGA/PVA-22), using a microfluidic synthesis method. The nanoparticle size distribution was measured as the hydrodynamic diameter by dynamic light scattering (DLS; Table 1), and the average diameter size of PLGA/PVA was 72.24 ± 0.7 nm, PLGA/PVA-21 was 60.15 ± 1.4 nm and PLGA/PVA-22 was 60.68 ± 1 nm, with an 85% encapsulation efficiency, and the fluorescent dye label coumarin-6 PLGA/PVA-C6 was 62.96 ± 0.7 nm. The morphology of NPs embedded in vitreous thin ice was revealed in the bright field cryo-TEM images. As shown in Figure 2, the NPs in formulation of PLGA/PVA, PLGA/PVA-21, and PLGA/PVA-22 are all spherical micelles and the sizes agree with the DLS size distribution measurement. The majority of NPs in SQ-encapsulated formulations are core-shell structured with an electron-dense core (Figure 2B,D); whereas the NPs in PLGA/PVA formulation are core-shell structured with less electron-dense or electron-lucent core (Figure 1A). The SQs were encapsulated in the core of PLGA/PVA NPs.

### 3.3. Effect of NPs on the Viability of LAD2 Cells

Although PLGA/PVA NPs did not trigger significant cytotoxicity, the PLGA/PVA-SQ combination could be cytotoxic to the human mast cells over longer periods of exposure. Consequently, the effect of PLGA/PVA-SQ NPs on the LAD2 viability was measured over a 72-h period (Figure 2D). No significant metabolic changes were observed on LAD2 cells exposed to NP after 1, 3, 48, or 72 h of treatment. However, the LAD2 cells treated with the NPs after 24 h showed slightly increased viability (~10%) compared to the control cells (Figure 2E). Cell toxicity was also evaluated by examining the gross morphology alterations of the nuclei normally associated with cell death (Figure 3F–M) and showed no observable differences compared to untreated control cells.

### 3.4. Internalization of NPs by LAD2 Cells

Previous studies using PLGA NPs have shown rapid nanoparticle internalization, but the kinetics of cellular internalization, intracellular distribution and nanoparticle retention are poorly understood, particularly for mast cells. PLGA/PVA NPs with encapsulated fluorescent compound coumarin-6 were used as a test for nanoparticle internalization.

Internalization of fluorescently labelled hybrid PLGA/PVA NPs (PLGA/PVA-C6) by LAD2 was analyzed by flow cytometry. LAD2 cells treated with PLGA/PVA-C6 showed increased fluorescence compared to untreated cells (Figure 3A). Furthermore, 24 h after treatment with PLGA/PVA the cells were not fluorescent but 97.5% of the LAD2 cells treated with PLGA/PVA-C6 were fluorescent (Figure 3B). Moreover, LAD2 cells exposed to PLGA/PVA-C6 for 3 h were fluorescent (arrow heads indicate cell-associated fluorescence; Figure 3C–E) as measured by fluorescence microscopy. 

TEM analysis of LAD2 cells treated with the NPs was performed (Figure 3F–M) to confirm internalization. TEM analysis showed that the NPs were rapidly internalized by the LAD2 cells after 3 h (Figure 3G–I—arrow heads shown the internalized NPs), and 24 h of treatment (Figure 3K–M). Furthermore, this internalization appears to be associated with specific intracellular structures such as granules and particle-containing cytoplasmic vesicles. At least two kinds of intracellular transport system for internalized PLGA nanoparticles have been suggested; one is mediated by a recycling endosome that turns over rapidly, and the other is a more complex vesicle transport system that turns over relatively slowly. When exposed for 2 h, most nanoparticles are internalized by the cells and are thought to be present in the recycling endosomes. In contrast, when cells are exposed for 20 h or more, a substantial fraction of nanoparticle-containing vesicles escape from the recycling endosome pathway and subsequently interact with the cytoplasmic compartments such as ER, Golgi apparatus, and secretory granules [17]. Whereas the intracellular structure of LAD2 cells showed no obvious changes between 3 h and 24 h incubation with NPs (as shown in Figure 3G–I,K–M) several PLGA/PVA-21 and PLGA/PVA-22 NPs were internalized and visible after 24 h.

### 3.5. Hybrid Copolymer NPs Modify Kit and MrgprX2 Gene Expression

Mast cells are activated via their surface receptors such as FcεRI and c-Kit. To determine whether PLGA/PVA hybrid NPs could potentially modify LAD2 cell responses via these receptors, we measured the gene expression of these genes by real time-PCR after NP treatment. PLGA/PVA NPs had no effect on the *FcεRI* subunit expression or *KIT* gene expression. PLGA/PVA-21 significantly decreased the expression of all subunits of *FcεRI* (*p* < 0.05) (Figure 4C) and *KIT* (*p* < 0.001) (Figure 4D) compared to untreated cells. However, *MrgprX2* expression was not affected after PLGA/PVA-21 treatment (Figure 4E). PLGA/PVA-22 showed statistical differences in the expression of *MrgprX2* (*p* < 0.001) compared to untreated, and PLGA/PVA treatments. The protein expression of FcεRIα was also assessed by changes in the median fluorescence intensity (MFI) of LAD2 cells exposed overnight to NP formulations and analyzed by flow cytometry; Figure 4F shows that PLGA/PVA-21 as well as PLGA/PVA-22 exhibit significant changes (*p* < 0.001, and *p* < 0.05, respectively) in the expression of FεRIα receptors, compared to untreated control cells, and PLGA/PVA-treated LAD2 cells. Flow cytometry analysis confirmed decreased FcεRIα and c-Kit expression, showing a significant decrease (*p* < 0.001) in protein expression after PLGA/PVA-21 treatment (Figure 4F, and Figure 4G, respectively). In addition to FcεRI, mast cells are also activated by the mas-related G protein-coupled receptor (MrpgrX2). All three NPs significantly increased the expression of MrgprX2, but PLGA/PVA-21 was the most potent, and increased MrgprX2 expression almost two-fold (Figure 4H). This difference was statistically significant (*p* < 0.001), compared to PLGA/PVA alone, suggesting that the increased effect was due to the SQ payload.

### 3.6. Effect of Hybrid Copolymer NPs on LAD2 Cell Granule Contents (Tryptase and Chymase Expression)

Tryptase and chymase are characteristic mast cell-specific proteases that are released upon mast cell activation and activate inflammation by stimulating neutrophil activation [18], tissue fibrosis [19], as well as stimulating the release of granulocyte chemoattractant IL-8 [20], and up-regulating the expression of ICAM-1 on the epithelial cells [21]. In addition, tryptase induces the expression of mRNA for IL-1β, which may be important for the recruitment of inflammatory cells to sites of mast cell activation [22]. Tryptase and chymase are synthesized de novo and stored in secretory granules, and changes in granule morphology are often associated with changes in tryptase and/or chymase expression (9). Therefore, to determine whether the hybrid NPs modified mast cell granule contents, we measured *TPSAB1* (tryptase) and *CMA1* (chymase) gene expression and protein expression by LAD2 cells exposed to the PLGA/PVA NPs. PLGA/PVA-21 and PLGA/PVA-22 significantly increased *TPSAB1* gene expression almost two-fold (*p* < 0.001) compared to untreated cells (Figure 5A), while none of the hybrid NPs had any significant effect on *CMA1* expression as measured by qPCR (Figure 5B).

Tryptase and chymase protein expression was measured by intracellular flow cytometry. In contrast to the qPCR data above, PLGA/PVA and PLGA/PVA-21 significantly decreased (*p* < 0.001) tryptase protein expression by 15% and 20%, respectively, compared to untreated cells (Figure 5C). PLGA/PVA-22 did not change intracellular tryptase expression. In contrast to the qPCR data, PLGA/PVA decreased chymase protein expression by approximately 30%, while PLGA/PVA-22 decreased chymase protein expression by almost 70% (*p* < 0.001). PLGA-PVA-21 increased chymase expression by 10% (Figure 5D).

### 3.7. Effect of Hybrid Copolymer NPs on Human Mast Cell Degranulation

Mast cell degranulation is a hallmark of mast cell activation and requires a complex set of energy-dependent movements of intracellular machinery, including the fusion of granules with the extracellular membrane [23,24,25]. To determine whether hybrid copolymer NPs influenced degranulation events, LAD2 cells were treated with PLGA/PVA, PLGA/PVA-21 (25 μM), or PLGA/PVA-22 (25 μM) for 30 min, then stimulated via their FcεRI receptors (with IgE-SA), and the release of the granule marker, β-hexosaminidase, was measured. PLGA/PVA and PLGA/PVA-21 decreased the degranulation of LAD cells in 25 ± 1.3% and 22 ± 1.6%, respectively (Figure 6A). PLGA/PVA-22 had no significant effect on LAD2 degranulation, but on its own, PLGA/PVA-22 increased a small amount of degranulation (18.5 ± 3.1%) compared to untreated cells (Figure 6B), suggesting that any effects of the PLGA/PVA are counteracted by SQ 22.

To determine the effects of free SQs on LAD2 degranulation, we measured FcεRI-stimulated LAD2 degranulation after treatment with SQs alone (in the absence of the PLGA/PVA carrier (Figure 6C). SQ21 inhibited degranulation (28 ± 2.7%) compared to IgE-SA-activated cells (39.8 ± 2.8%). SQ22 appeared to slightly decrease degranulation but not reach statistical significance compared to IgE-SA-activated cells (32.8 ± 2.5%, Figure 6C). Finally, the effect of SQs on resting cell degranulation was measured. Neither SQ21 nor SQ22 had any measurable effect on resting cell degranulation (Figure 6D).

## 4. Discussion

Overall, our results suggest that PLGA/PVA-21 and PLGA/PVA-22 alter human mast cell phenotypes and inhibit their activation without modifying viability. While free SQ21 and SQ22 decreased LAD2 cell viability by 20%, PLGA/PVA-21 and PLGA/PVA-22 did not alter LAD2 viability, even after a 72-h treatment, supporting the observation that encapsulation improves the biocompatibility of SQ. Since SQ can be very cytotoxic to cells, particularly in in vitro experiments, the PLGA/PVA encapsulation represents a significant improvement in SQ delivery.

Butterbur (*Petasites japonicus*) is a herbaceous perennial plant belonging to the family Asteraceae, native to Europe and northern Asia [26]. Sesquiterpenoids, especially eremophilanes, have been previously reported as the main constituents of this plant [27,28,29]. Previous studies have suggested that *P. japonicus* extracts possessed a variety of biological features such as neuroprotective [30,31,32], anti-allergic [33,34,35], and anti-inflammatory properties [35,36]. More recent studies from our group demonstrated that lipophilic eremophilane-type SQs fukinone (SQ21), and 10βH-8α,12-epidioxyeremophil-7(11)-en-8β-ol (SQ22), isolated and purified from rhizome of *Petasites tatewakianus*, modify dendritic cell function, inhibiting the maturation and activation of bone marrow-derived dendritic cells [12,13].

Nanomaterials such as PLGA have been widely used as drug delivery vehicles due to their biocompatibility and biodegradability. Many polymer-based nanomaterials have a short half-life and exhibit poor absorption by the gastrointestinal tract, and are highly susceptible to metabolic degradation in the liver. However, PLGA nanomaterials do not have these disadvantages. 

Exposure to engineered sometimes activates immune responses including inflammation, hypersensitivity, and immunosuppression. Engineered nanoparticles interact with the immune system through key effector cells (i.e., mast cells and antigen-presenting cells) or via complement activation, where they modify the immune cell and complement responses. Innate immune cells recognize danger or damage-associated molecular patterns (DAMPS or ‘alarmins’) such as interleukin-1 (IL-1), which, in turn, trigger the inflammatory response; some nanoparticles act as danger signals, termed nanomaterial-associated molecular patterns (NAMPs), promoting cellular stress responses, cell injury, and inflammation [37]. Several studies have reported that NALP3 (also known as NLRP3 and cryopyrin) mediates the formation of caspase-1, activating the multiprotein complex known as the inflammasome [38]. Size-dependent activation of innate immune signaling pathways has also been shown. Small (<10 nm) gold (Au) nanoparticles preferentially activate the NLRP3 inflammasome and induce caspase-1 maturation and IL-1 production, while larger Au nanoparticles (>10 nm) trigger the NFκB signaling pathway [39]. Polymer-based, or cell-penetrating nanoparticles activate inflammasome-enhancing antibody production in response to nanoparticle exposure. In this context, Demento et al. (2009) reported that macrophage exposure to LPS-modified antigen-loaded PLGA nanoparticles resulted in an effective vector vaccine via both TLR and inflammasome activation, in which PLGA nanoparticles allowed for the targeted delivery, protection, and sustained release of antigen during vaccination. Their studies demonstrated that LPS-modified antigen loaded PLGA nanoparticles effectively entered the antigen-presenting cells and elicited both humoral and cellular immunity against encapsulated antigens in mice. This demonstrated that PLGA nanoparticles are useful in vaccine development, inducing protection in a murine model of west Nile encephalitis [40]. Additional studies have demonstrated that gold nanoparticles bind to fibrinogen, promoting the activation of Mac-1 receptor and subsequent NFκB-signaling in macrophages [41].

Mast cells are innate effector cells, acting as the first line of defense against invading pathogens and foreign antigens. Mast cells also mediate allergic and inflammatory responses and influence neovascularization and tissue remodeling. Mast cells are found in all tissues, particularly those associated with vasculature and nerves, as well as near epithelial tissues such as airways and skin. Mast cells are also critical effector cells in the adaptive immune response mediated by immunoglobulin type E (IgE). In this study, we have demonstrated that human mast cells can internalize the biodegradable hybrid copolymer nanoparticles PLGA/PVA-SQ21 and -SQ22, and this internalization modified the expression of several inflammatory mediators such as *Kit* and *MrgprX2*, as well as mast cell-specific tryptase. PLGA/PVA-21 and PLGA/PVA-22 inhibited tryptase expression but had no effect on chymase expression, thereby promoting a shift to the tryptase-positive phenotype (MC_T_). Mast cells are subdivided into two main phenotypic categories based upon their expression of tryptase and chymase, such that mast cells positive for both tryptase and chymase (MC_CT_) are often present at high levels in connective tissues such as the skin, whereas mast cells that express primarily tryptase (MC_T_) are present at high numbers in mucosal tissues including the gut and lung (17). MC_CT_ are also the only mast cell phenotype to express MrgprX2, suggesting that the PLGA/PVA-22 are fundamentally altering the mast cell phenotype.

PLGA/PVA-21 and PLGA/PVA-22 decreased expression of the subunits of the high affinity IgE receptor (*FcεR1α*, *FcεR1β*, *FcεR1γ*, *Kit*, suggesting that hybrid NPs could alter mast cell responses to antigen which binds to cell surface IgE and crosslinks FcεRI to initiate the release of pro-inflammatory mediators. Consistently, PLGA/PVA-21 and PLGA/PVA-22 inhibited mast cell degranulation when the LAD2 cells were activated via the high-affinity IgE receptor (FcεRI), implying that the decrease in FcεRI expression could be responsible for the decrease in degranulation. Mast-cell survival and differentiation requires stem cell factor (SCF) and its ligand c-KIT receptor; thus, the SCF/c-KIT interaction is critical for mast-cell function, facilitating the secretion of pro-inflammatory mediators including cytokines (i.e., IL-4, IL-13, TNF, etc.), and chemokines. The correlation between the SCF/c-KIT pathway and the intensity of the acute inflammatory response is proportional to mast cell function and degranulation. Previous reports have shown that the tyrosine kinase inhibitors masitinib and imatinib mesylate effectively inhibit c-KIT receptor tyrosine kinase signaling activity [42,43,44]. Avula et al.’s (2013) study of a murine subcutaneous implant model showed that PLGA nanoparticles loaded with the tyrosine kinase inhibitor masitinib inhibited tissue resident mast cells and modified the intensity of inflammatory cell recruitment to implant sites compared to controls [45].

Previous studies by our group have shown that SQs inhibit maturation and activation of dendritic cells in a process that is augmented when peroxisome proliferator-activated receptor gamma (PPARγ) is activated. Our data has now demonstrated that SQ-loaded PLGA/PVA hybrid copolymer nanoparticles also modify human mast cells, altering their granule contents and inhibiting degranulation. Our results suggest that SQs act as PPARγ ligands, mediating mast cell anti-inflammatory effects, modulating the expression of co-stimulatory molecules, altering their phenotype, and leading to an impaired expression of pro-inflammatory factors involved in the inflammatory response. A schematic representation of PLGA/PVA-21 and -22 NPs is presented in Figure 7. We propose that PLGA/PVA-21 or -22 nanoparticles are internalized through the plasma membrane using an unknown mechanism, and move into the cytoplasm **1**. Cellular internalization of hybrid copolymer nanoparticles could use several endocytic pathways **2**, wherein the nanoparticles’ physicochemical properties might have significant effects on their internalization. Lysosomal processing of the PLGA/PVA structure could degrade the nanoparticles and release SQs; some SQs could directly bind to mast cell granules and inhibit tryptase, histamine, and β-hexosaminidase release **3**. SQ may also bind PPARγ **4**, and activate PPARγ to migrate into the nucleus **5**, heterodimerizing with its obligate partner, the retinoic acid-X receptor (RXR). The PPARγ-RXR complex could bind to the peroxisome proliferator response elements (PPREs) in the promoter region of target genes in distinct regions of DNA, resulting in decreased expression of *FcεR1α*, *FcεR1β*, *FcεR1γ*, and *Kit*. Indirect effects could eventually lead to decreased expression of tryptase and morphological changes.

## 5. Conclusions

Although sesquiterpenes are often considered to be anti-inflammatory molecules, they are often cytotoxic and not biocompatible with all tissues. It is possible to utilize hybrid polymer-based NPs to encapsulate such molecules to render them more readily accessible to cells in culture. SQ-loaded hybrid nanoparticles PLGA/PVA-21 and PLGA/PVA-22 were synthesized using a microfluidic approach, generating particles of 60 to 70 nm that were efficiently internalized by human mast cells. Similarly, internalized SQ-loaded NPs did not alter human mast cell viability, in contrast to free SQ, which decreased LAD2 cell viability by approximately 20%. The SQ-loaded NPs also had potentially anti-inflammatory properties since they decreased the expression of inflammatory receptors (*FcεR1α*, *FcεR1β*, *FcεR1γ*, and *Kit*), inhibited the expression of tryptase and decreased degranulation when the cells were activated via FcεRI. Overall, our results suggest that the strategy of encapsulating SQ into a hybrid PLGA/PVA NP using a microfluidic approach is feasible and results in NPs that are both biocompatible and effective inhibitors of human mast cell activation, in the context of an IgE-mediated response. This observation could have meaningful implications for the design of similar anti-inflammatory approaches.

## Figures and Tables

**Figure 1 nanomaterials-11-00953-f001:**
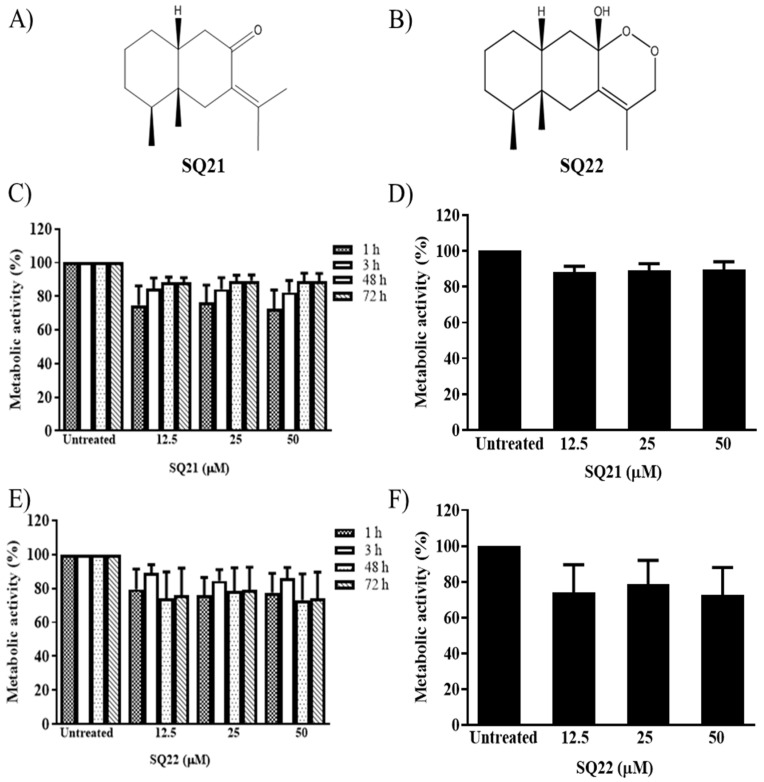
Chemical structure of fukinone (SQ21, (**A**)), and 10βH-8α, 12-Epidioxyeremophil-7(11)-en-8β-ol (SQ22, (**B**)). Human mast cells (LAD2 cells) were treated with 12.5, 25, or 50 μM of the bicyclic eremophilane SQ21 (**C**,**D**), and SQ22 (**E**,**F**), and cell viability was evaluated 1 to 72 h after treatment, and expressed as percentage of metabolic activity rate with respect to untreated control cells (*n* = 3, ±SEM). LAD2 cells treated for 24 h with SQ21 (**D**), or SQ22 (**F**) are shown separately.

**Figure 2 nanomaterials-11-00953-f002:**
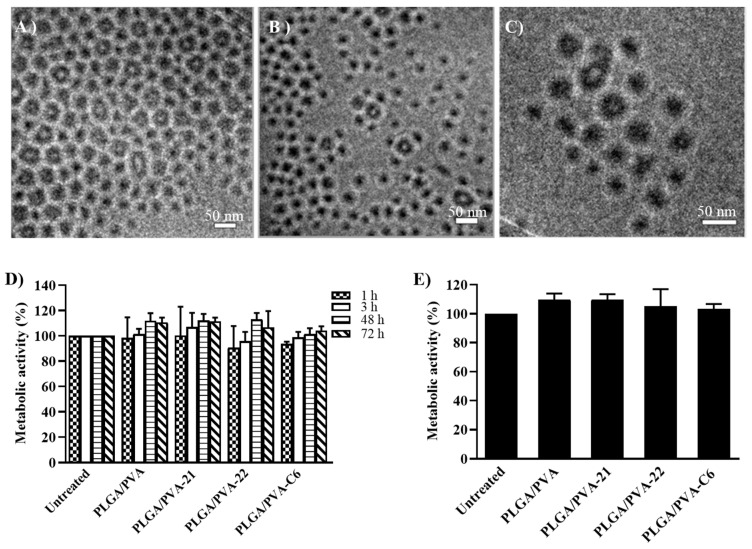
Characterization of PLGA/PVA SQ NPs by cryoTEM is shown for PLGA/PVA (**A**), PLGA/PVA-21 (**B**), and PLGA/PVA-22 (**C**). Scale bars are 50 nm. Cell viability measured by percentage of metabolic activity of LAD2 cells exposed to SQ NPs PLGA/PVA-21 and PLGA/PVA-22, in comparison to untreated controls, were determined 1, 3, 48, and 72 h after treatment, and analyzed by the XTT assay ((**D**), *n* = 3, ±SEM). LAD2 cells treated with PLGA/PVA NPs after 24 h treatment (**E**).

**Figure 3 nanomaterials-11-00953-f003:**
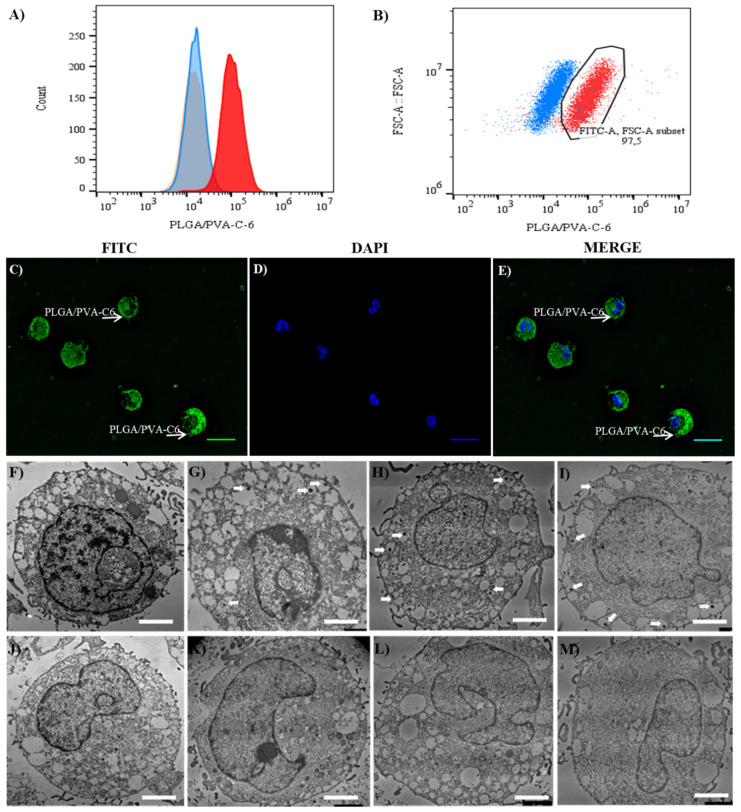
PLGA/PVA nanoparticle internalization. Panel (**A**) shows the cell internalization of PLGA/PVA-C6, determined by flow cytometry analysis, histograms in panel (**A**) show a shift in fluorescence intensity of LAD2 cells exposed to PLGA/PVA-C6 for 24 h (red histogram), compared to untreated control cells (orange histogram) and PLGA/PVA (blue histogram) controls. Percentages of fluorescently labelled LAD2 cell-gated population in comparison to PLGA/PVA treated cells are also shown in panel (**B**). Cell internalization of green fluorescent coumarin-6-loaded NPs visualized using an FITC filter by fluorescence microscopy, showing cell internalization by LAD2 cells (**C**), DAPI-stained nuclei (**D**), and merge of PLGA/PVA-C6 and DAPI stains (**E**), scale bar 10 µm. TEM micrographs of intracellular structure of LAD2 cells exposed to PLGA/PVA-21 (**H**,**L**), PLGA/PVA-22 (**I**,**M**), and PLGA/PVA (**G**,**K**). Untreated cells (**F**,**J**) were included as controls. LAD2 cells were exposed to PLGA/PVA SQ nanoparticles for 3 h (**F**–**I**), and 24 h (**J**–**M**). Scale bars are 2 µm, arrowheads show internalized PLGA/PVA nanoparticles.

**Figure 4 nanomaterials-11-00953-f004:**
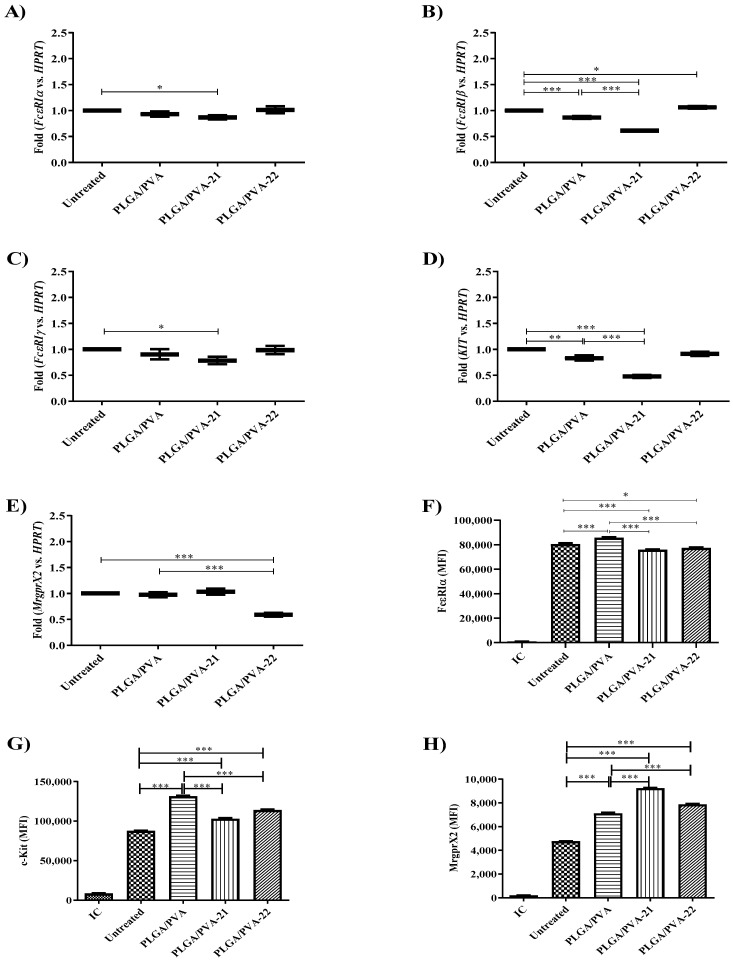
Effect of SQ nanoparticle administration to LAD2 cells on receptor expression. LAD2 cells were exposed for 6 h to PLGA/PVA-21, or PLGA/PVA-22, followed by RNA isolation, and cDNA synthesis. Gene expression levels of *FcεRIα* (**A**), *FcεRIβ* (**B**), *FcεRIγ* (**C**), *Kit* (**D**), and *MrgprX2* (**E**) were analyzed by real-time PCR, using *HPRT1* to normalize gene expression (*n* = 3 ± SEM; ** p* ≤ 0.05, ** *p* ≤ 0.01, **** p* ≤ 0.001). FcεRIα (**F**), c-Kit (**G**), and MrgprX2 (**H**) protein synthesis by LAD2 cells treated for 24 h with PLGA/PVA-21, or PLGA/PVA-22 were analyzed by flow cytometry. Untreated and PLGA/PVA-treated cells were included as controls in all tested conditions. (*n* = 3 ± SEM; * *p* ≤ 0.05, ** *p* ≤ 0.01, *** *p* ≤ 0.001).

**Figure 5 nanomaterials-11-00953-f005:**
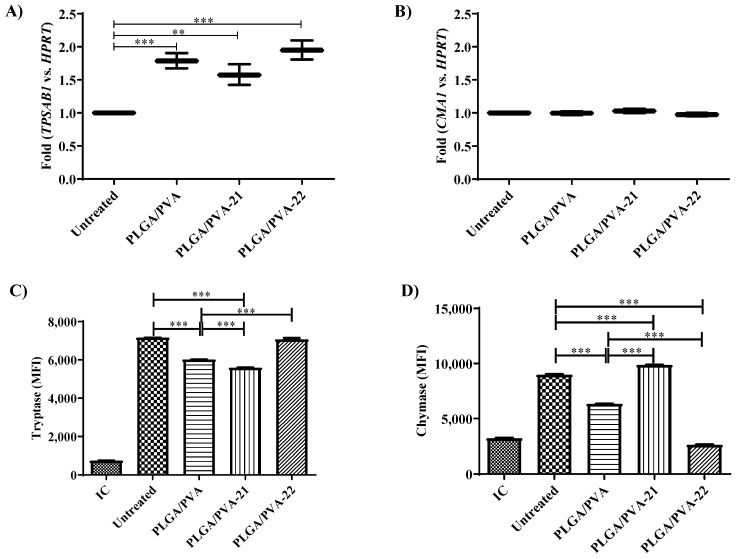
Effect of hybrid copolymer NPs on LAD2 cell granule contents. LAD2 cells were exposed for 6 h to PLGA/PVA-21, or PLGA/PVA-22, followed by RNA isolation, and cDNA synthesis. Gene expression levels of *TPSAB1* (tryptase, (**A**)) or *CMA* (chymase, (**B**)) were analyzed by real-time PCR using *HPRT1* to normalize gene expression (*n* = 3 ± SEM; * *p* ≤ 0.05, ** *p* ≤ 0.01, *** *p* ≤ 0.001). Intracellular tryptase (**C**) and chymase (**D**) synthesis was analyzed on LAD2 cells treated for 24 h with PLGA/PVA-21 or PLGA/PVA-22 by flow cytometry. Untreated and PLGA/PVA-treated cells were included as controls in all tested conditions. (*n* = 3 ± SEM; * *p* ≤ 0.05, ** *p* ≤ 0.01, *** *p* ≤ 0.001).

**Figure 6 nanomaterials-11-00953-f006:**
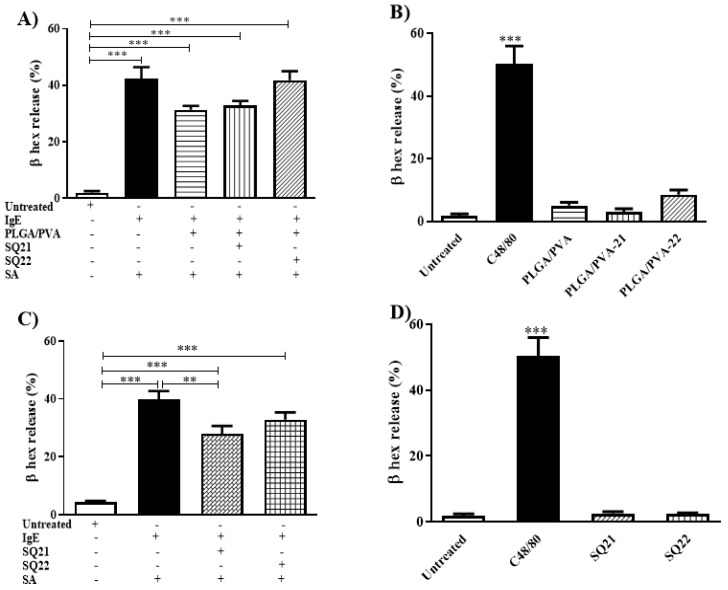
Inflammation mediator released by LAD2 cells treated with encapsulated SQs. LAD2 cells were sensitized overnight with biotinylated IgE, exposed to PLGA/PVA-21 or PLGA/PVA-22 (**A**), or SQ21, SQ22 (**C**) for 3 h and challenged with streptavidin for 30 min, and β-hexosaminidase release was evaluated in LAD2 cells. Untreated cells or IgE-SA sensitized and challenged cells were included as negative and positive controls, respectively. PLGA/PVA-21 or PLGA/PVA-22 (**B**), or SQ21 and SQ22 (**D**) cell activation was measured after 30 min incubation. Untreated and compound 48/80 treated cells were included as negative and positive controls, respectively (*n* = 3 ± SEM; ** *p* ≤ 0.01, *** *p* ≤ 0.001).

**Figure 7 nanomaterials-11-00953-f007:**
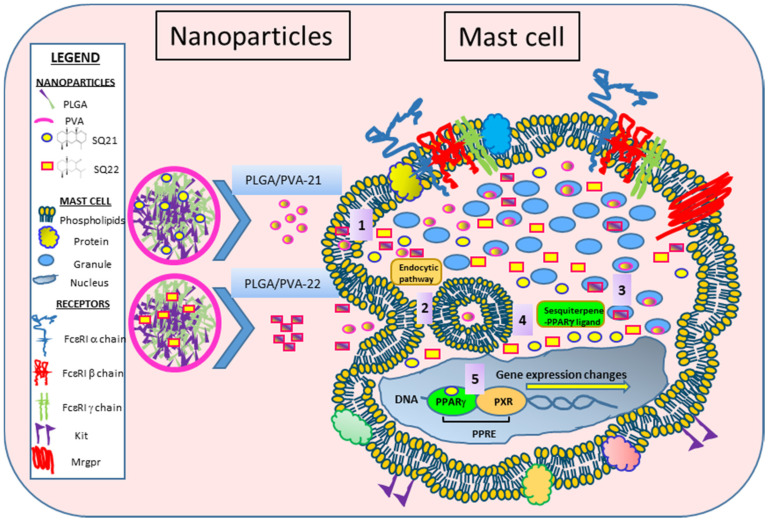
PLGA/PVA-hybrid SQ nanoparticles and cell internalization model. 1. PLGA/PVA-21 or -22 nanoparticles travel directly through the lipid bilayer of the cell membrane. 2. Nanoparticles are taken up by the endocytic pathway. 3. PLGA/PVA is hydrolyzed and SQs are released into the cell cytoplasm as PPARγ ligands. 4. SQs bind to cell granules. 5. SQs bind to the PPARγ transcription factor in the nucleus, which, in turn, binds to the retinoic acid-X receptor (RXR), forming the PPRE complex on the DNA and modifies gene expression.

**Table 1 nanomaterials-11-00953-t001:** PLGA/PVA nanoparticle formulations and particle characterization measured by dynamic light scattering.

Nanoparticle ID	Composition	Dh (nm)	PDI
**PLGA/PVA**	PLGA + PVA	72.24 ± 0.7	0.14
**PLGA/PVA-21**	PLGA + PVA + sesquiterepene21	60.15 ± 1.4	0.11
**PLGA/PVA-22**	PLGA + PVA + sesquiterpene 22	60.68 ± 1	0.08
**PLGA/PVA-C6**	PLGA + PVA + coumarin 6	62.96 ± 0.7	0.13

Poly D, L-lactide-co-glicolide 5 mg/mL in acetonitrile/PVA 2.0% in water (PLGA/PVA), PLGA/PVA/fukinone (PLGA/PVA/21), PLGA/PVA/10βH-8α,12-epidioxyeremophil-7(11)-en-8β-ol (PLGA/PVA/22), PLGA/PVA/Coumarin 6 (PLGA/PVA-C6). Dh: particle size as hydrodynamic diameter (Dh, *n* = 3 ± SEM). PDI: polydispersity index.

**Table 2 nanomaterials-11-00953-t002:** Primer and probe sequences used for real-time PCR analysis.

ID	Sequence
**FcεR1α, Homo sapiens**	
primer	GCA AAC AGA ATC ACC ACC AAC
primer	GTT GAA GAC AGT GGA ACC TAC T
probe	/5HEX/CTC AGA CTC/ZEN/ATA GTC CAG CTG CCA C/3IABkFQ/
**FcεR1β, Homo sapiens**	
primer	TCT TCA TAA ACA CGA TCC TCT GG
primer	GAT GCT GTT TCT CAC CAT TCT G
probe	/5HEX/TTG AGT TCT/ZEN/TCC CCA GCT CCA CAG/3IABkFQ/
**FcεR1γ, Homo sapiens**	
primer	CTC ATG CTT CAG AGT CTC GTA
primer	GAC TGA AGA TCC AAG TGC GAA
probe	/56-FAM/TGG TGC TCA GGC CCG TGT AAA C/36-TAMSp/
**Kit, Homo sapiens**	
primer	TCA GTG CAT AAC AGC CTA ATC TC
primer	GTT CTG CTC CTA CTG CTT CG
probe	/56-FAM/TTC CCC TGG/ZEN/ACT CAC AGA TGG TTG/3IABkFQ/
**MrgprX2, Homo sapiens**	
primer	GGA CTG AGA AAG TTC AGC AAA TC
primer	GAA GCA GTC TGG TGT AGA GAT G
probe	/56-FAM/TGT GGC TTT GAG AGG CAA CTT TGC/36-TAMSp/
**Tryptase, Homo sapiens**	
primer	CAG TGG TGT TTT GGA CAG C
primer	CGG CCT GGC ATC TAC AC
probe	/56-FAM/TGA CTC ACG/ZEN/GCT TTT TGG GGA CAT/3IABkFQ/
**Chymase, Homo sapiens**	
primer	CGT GGT GAA GAG TAG AAG TGT T
primer	GTC TAT AAC AGT CAC CCT TGG A
probe	/5HEX/AGA GGA AGA/ZEN/AGA CAC ATG GCA GAA GC/3IABkFQ/
**HPRT1, Homo sapiens**	
primer	GCG ATG TCA ATA GGA CTC CAG
primer	TTG TTG TAG GAT ATG CCC TTG A
probe	/5HEX/AGC CTA AGA/ZEN/TGA GAG TTC AAG TTG AGT TTG G/3IABkFQ/ Or /56-FAM/AGC CTA AGA/ZEN/TGA GAG TTC AAG TTG AGT TTG G/3IABkFQ/

## Data Availability

The data that support the findings of this study are available on request from the corresponding author.

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
