# Peer review of "Sesquiterpene-Loaded Co-Polymer Hybrid Nanoparticle Effects on Human Mast Cell Surface Receptor Expression, Granule Contents, and Degranulation"

_nanomaterials, 2021, doi:10.3390/nano11040953_

Round 1

Reviewer 1 Report

The paper is well written and organised. The topic is of interest.

Author Response

We thank the reviewer for the supportive comments.

Reviewer 2 Report

The main idea of the article seems to be interesting. In general, paper is well-organized and properly prepared. Discussion over the results of the experiments is proper and extensive. Proposed work is worth appreciating and considering for publication in the journal but some elements of the article need to be improved – all suggestions are given in more detail below.

  • Title of the paper should be changed and not expressed as a conclusion but as a brief statement showing the main concept of the paper.
  • Notation [3-9] is not adequate. Authors need to divide such a range into smaller compartments and then comment them briefly.
  • Section 2.2.: Processes of isolation and purification of the bicyclic compounds need to be characterized in few sentences.
  • Table 1: the table should be supplemented with additional table row with the name of the parameters which values are presented (i.e. PDI and hydrodynamic diameter). Its caption should be brief and the names of parameters should be indicated in the table and not in its caption.
  • Section 2.5.: the principle of XTT test should be briefly described.
  • Statement such as “Nanomaterials such as Polylactic-co-glycolic acid (PLGA) (…)” is not proper and should be corrected (second paragraph in section Discussion). Additionally, this sentences should be re-written because “drug delivery systems ap-proved by the FDA and European Agencies” or “parenteral use for drug administration” are not properties.
  • Final paragraph of Section 4. should be developed and presented as Section 5. Conclusions.
  • Quality of Figs 2-3 should be improved.

Author Response

  1. We thank the reviewer for the suggestion to change the title. The title is a reflection of the contents of the manuscript which is the examination of sesquiterpene-loaded hybrid nanoparticles and their effects on human mast cell functions such as degranulation and receptor expression. In keeping with similar titles from some of our colleagues published in other journals, we feel that this title best encapsulates the focus of the research.
  2. We agree that the original references in the introduction were listed together and were not adequately addressed in the manuscript. As a result, we have removed some of the refernces and focused on others that directly address our argument. (p. 3)
  3. The reviewer asks for a more thorough explanation of the processes used for the isolation and purification of the bicyclic compounds. We have now included a more detailed explanation of the methods used to isolate the compunds on page 4.
  4. The reviewer has asked for more information (PDI and hydrodynamic diameter) in Table 1. This information has now been added to the table.
  5. The reviewer asked for the theory behind the common and extensively used XTT assay. We have now added some information about the theory behind the XTT assay, which is very similar to the MTT assay, on page 15.
  6. The reviewer asked for a modification of the statement "Nanomaterials such as PLGA". The reviewer is correct that PLGA is not, in of itself, a nanomaterial and therefore we have corrected any such statements throughout.
  7. The reviewer asks for an extension of the final paragraph in the discussion section to present some conlucions. We have now expanded this section with some conclusive statements.
  8. The reviewer asks for improved Fig. 2-3. We have re-done the EM analysis of the NPs and replaced Fig. 2 and Fig. 3 with better quality images. The NPs were embedded in vitreous ice and these were well dispersed as many particles are present in the field of view. Part of the figure in panel A (near the center) and B (on the upper right) has a brighter contrast which is a common phenomenon due to charging of the ice under the electron beam. The low beam dose imaging mode is also used to mitigate the radiation damage for such soft material. In general, the quality of these cryoTEM images is much improved and of high quality. 

Reviewer 3 Report

The manuscript by Arizmendi et al. describes synthesis and effects of sesquiterpene-loaded co-polymer hybrid nanoparticles on mast cells. Description of the methodology and presentation of the results are clear. There is, on the other hand, information on the advantage/use of the formulation missing and, therefore, it is difficult to assess importance of the findings.

General comments

There is no explanation on the planned use and administration of the particles. It is also not stated if encapsulation resulted in higher efficacy than the exposure to the non-encapsulated sesquiterpenes.

Specific comments

-Physicochemical characterization of the particles in complete media as applied to the cells is missing.

-It is mentioned that the PLGA/PVA particles did not display auto-fluorescence. Did the cells have any auto-fluorescence when imaged at the same settings used for the particle uptakes in Fig. 3?

-morphology of most cells shown in Figures 3F)-M) is poor and identification of PLGA particles difficult because structures with similar density are visible also in the controls. Better examples should be provided.

Minor

- following information has to be added:

provider of the LAD2 cells

LOQ, LOD of detection

Dilutions of antibodies

Author Response

We thank the reviewer for the comments and have specific responses below.

  1. The reviewer asks for more detail as to the planned use and administration of the particles used in our study. They also ask whether the encapsulation resulted in a higher efficacy than the exposure to the non-encapuslated sesquiterpene. We show that SQ21 and SQ22 were successfully encapsulated by the PLGA/PVA hybrid structure and that this encapsulation faciliated internalization of the NPs and inhibited receptor expression. Encapsulation does not appear to influence degranulation as shown in Fig. 6, but likely results in more efficient internalization and bioavailability of the highly lipophilic compounds. Furthermore, the viability data in Fig. 1 and 2 indicates that the SQ compounds on their own decrease cell viability by approximately 20% even after 1 hr of exposure to the SQ, but when the SQ are encapsulated in the NP, the viability of the cells is still around 100%, even after 72 hr. We have added this observation to the abstract as well as the conclusions. Furthermore, we have rewritten the abstract, parts of the introduction and included a new conclusions section.
  2. The reviewer points out that the phytochemical characterizations of the particles in complete media is missing. The information has now been included on page 6.
  3. The reviewer asks of the cells or PLGA/PVA particles were auto-fluorescent. The same settings were used for all of the experiments performed using flow cytometry and neither the cells, nor the nanoparticles displayed autofluoresence in any of the detection filters commonly used in flow cytometry (with four excitation band filters; 355, 488, 638, and 808 nm). Similarly, the same settings were used when imaging using fluorescence microscopy. The gain and the sensitivity of the filters were adjusted such that the cells were not autofluorescent, nor the empty PLGA/PVA nanoparticles.
  4. The reviewer points out that the morphology of the cells in Fig. 3 is poor. As such, we have now included a new figure with improved image quality.
  5. The source of the LAD2 cells has been included in the Materials and Methods section.
  6. LOQ and LOD has been added to page 6.
  7. The dilutions of the antibodies has now been added to page 7.

Round 2

Reviewer 3 Report

The authors have addressed my comments.

Author Response

We have no gone through and thoroughly edited the manuscript for minor spelling and grammar errors.